# Decoding Susceptibility to Respiratory Viral Infections and Asthma Inception in Children

**DOI:** 10.3390/ijms21176372

**Published:** 2020-09-02

**Authors:** James F. Read, Anthony Bosco

**Affiliations:** 1Telethon Kids Institute, University of Western Australia, Nedlands, WA 6009, Australia; james.read@telethonkids.org.au; 2School of Medicine, The University of Western Australia, Perth, WA 6009, Australia

**Keywords:** Human Rhinovirus, respiratory syncytial virus, systems biology, omics, multi-omics, asthma, wheeze, innate immunity, Bow-tie architecture, early life

## Abstract

Human Respiratory Syncytial Virus and Human Rhinovirus are the most frequent cause of respiratory tract infections in infants and children and are major triggers of acute viral bronchiolitis, wheezing and asthma exacerbations. Here, we will discuss the application of the powerful tools of systems biology to decode the molecular mechanisms that determine risk for infection and subsequent asthma. An important conceptual advance is the understanding that the innate immune system is governed by a Bow-tie architecture, where diverse input signals converge onto a few core pathways (e.g., IRF7), which in turn generate diverse outputs that orchestrate effector and regulatory functions. Molecular profiling studies in children with severe exacerbations of asthma/wheeze have identified two major immunological phenotypes. The IRF7hi phenotype is characterised by robust upregulation of antiviral response networks, and the IRF7lo phenotype is characterised by upregulation of markers of TGFβ signalling and type 2 inflammation. Similar phenotypes have been identified in infants and children with severe viral bronchiolitis. Notably, genome-wide association studies supported by experimental validation have identified key pathways that increase susceptibility to HRV infection (ORMDL3 and CHDR3) and modulate TGFβ signalling (GSDMB, TGFBR1, and SMAD3). Moreover, functional deficiencies in the activation of type I and III interferon responses are already evident at birth in children at risk of developing febrile lower respiratory tract infections and persistent asthma/wheeze, suggesting that the trajectory to asthma begins at birth or in utero. Finally, exposure to microbes and their products reprograms innate immunity and provides protection from the development of allergies and asthma in children, and therefore microbial products are logical candidates for the primary prevention of asthma.

## 1. Introduction

Human Respiratory Syncytial Virus (HRSV) and Human Rhinovirus (HRV) are the leading cause of respiratory tract infections in infants and children [1,2]. Most children experience mild or asymptomatic infection. However, in susceptible individuals, HRSV and HRV infections may result in severe disease [3]. HRSV reaches almost universal infection by age 2 and is the major causative agent of bronchiolitis in infants [4]. HRV and HRV-C species, in particular, are major triggers for severe wheezing and asthma exacerbations in children, accounting for at least two-thirds of episodes [5,6,7]. Bronchiolitis is the leading cause of hospitalisation for infants (<2 years) in the United States, contributing 17% of all infant hospitalisations [8,9]. Moreover, it is estimated that HRSV is responsible for more than 500,000 emergency department (ED) visits and 1.5 million outpatient visits among young children (<5 years) annually in the US [10]. There are more than 5.5 million asthmatic children in the US, and asthma exacerbations account for more than 550,000 ED visits, 100,000 hospitalisations and 4 million GP visits [11]. In addition to the burden of HRSV- and HRV-induced acute illness on health care resources, severe illnesses with either of these viruses are strongly linked to the development of persistent wheeze and asthma [5,12,13]. What remains an open question is whether or not these viral agents drive the development of the various clinical manifestations of asthma and wheeze (e.g., atopic asthma, non-atopic asthma, and transient/persistent/recurrent wheezing) in susceptible individuals or alternatively unmask a pre-existing susceptibility in children who were already on a trajectory towards asthma. Whilst both HRSV and HRV induce wheezing, HRV wheezing is a much stronger predictor of subsequent asthma than HRSV wheezing, especially in children with aeroallergen sensitisation [12]. Moreover, prevention of RSV infection in pre-term or high-risk infants with the monoclonal antibody palivizumab decreased recurrent wheezing but had no effect on atopic asthma or lung function [14,15,16,17]. Oral prednisolone treatment of first-time wheezers with HRV decreased time to recurrence in the subgroup of children with high viral loads [18,19]. Together, these data suggest that HRSV is a risk factor for non-atopic wheeze/asthma whereas HRV drives atopic asthma. Given that asthma is a highly complex and heterogeneous disease, untangling the role of these viruses in the development of specific asthma phenotypes will require a detailed understanding of the underlying cellular and molecular mechanisms.

Omics enables the systematic investigation of the molecular states that underpin phenotypic states. When combined with the powerful tools of systems biology, which model genes as components of an interconnected system, it is possible to unveil the intervening biology that connects interactions between genes and environmental exposures with expression of disease for hypothesis generation, detection of biomarkers and endotypes, and prioritization of therapeutic targets [20,21,22]. As biological processes are under constant regulatory control [23], a comprehensive understanding of health and disease will require an integrative perspective of how molecular features across multiple regulatory elements (DNA sequence variation, open chromatin states, transcription factor binding, transcription of mRNA, protein synthesis, and metabolites) work together to jointly contribute to emergent phenotypic states.

## 2. Genetics

The development of genotyping microarrays and next generation sequencing technologies has enabled the systematic investigation of associations between DNA sequence variations and disease in human populations. The first genome-wide association study (GWAS) for asthma identified the 17q12-21 region, which is now the most significant and replicated susceptibility locus associated with asthma in the genome [24,25]. The 17q locus comprises 17 asthma-associated SNPs, which span multiple candidate genes located in three regions divided into the core *(IKZF3, ZPBP2, GSDMB,* and *ORMDL3),* distal *(GSDMA)* and proximal regions *(ERBB2* and *PGAP3).* The core region is strongly associated with early onset asthma [26], and the proximal region is associated with asthma in adults [27,28]. Caliskan et al. [29] investigated the relationship between 17q genotypes, virus-induced wheezing and asthma risk in early life. They found that the 17q genotypes were associated with HRV wheezing but not HRSV wheezing, and additionally that the association of these variants with asthma was only observed in those children who experienced HRV wheezing [29]. These findings demonstrate that there is a specific interaction between 17q variants, HRV wheezing, and asthma risk in children. Interestingly, studies from children raised in farming environments have demonstrated that the same 17q variants that are associated with risk for wheezing and asthma are also associated with protection from these conditions in children exposed to animal sheds [30,31]. This suggests the hypothesis that 17q variants increase susceptibility to the effects of HRV on asthma risk in the absence of protective microbial exposures [28].

Decoding the role of the 17q region in asthma pathogenesis is extremely challenging because the locus harbors multiple candidate genes and is characterised by strong patterns of linkage disequilibrium in European and Asian populations [25]. To address this issue, previous studies have focused on patterns of gene expression. ORMDL3 and GSDMB are promising candidates in this regard because expression levels of these genes are highly correlated with asthma risk genotypes at 17q in blood cells and are also induced by HRV infection [29]. An alternative and elegant approach to pinpoint the putative causal genes is to study populations of African ancestry [32], where patterns of linkage disequilibrium are less pronounced. Employing this approach, Ober and colleagues reported that only two SNPs in the 17q region were associated with asthma in African American children, and these SNPs were correlated with expression of GSDMB but not ORMDL3 in airway epithelial cells [33]. The 17q variants that correlated with ORMDL3 and GSDMB in blood were not associated with asthma in African Americans, suggesting that these SNPs were not central to asthma pathogenesis, and by inference, GSDMB is the leading candidate gene in the region.

Functional studies have also been employed to dissect the role of candidate genes on 17q in asthma. Knockdown of ORMDL3 in airway epithelial cells resulted in a marked reduction in expression of ICAM1, which is the main receptor exploited by some HRV-A and HRV-B species for host cell entry, demonstrating a plausible functional link between ORMDL3 and HRV wheezing [34]. Liu et al. [35] demonstrated that silencing ORMDL3 reduced replication of HRV-16 in airway epithelial and HeLa cells following in vitro infection. Unlike the previous report, however, ICAM1 was not reduced following ORMDL3 knockdown which may suggest the influence of ORMDL3 occurs after viral binding [34,35]. ORMDL3 is also important in the regulation of endoplasmic reticulum stress and the unfolded protein response, and the regulation of levels of ceramide and sphingosine 1-sulfate [36,37]. GSDMB is highly expressed in bronchial epithelial cells from subjects with asthma and expression is correlated with disease severity. Overexpression of GSDMB in primary airway epithelial cells of human GSDMB transgenic mice induced 5-lipoxygenase and transforming growth factor-β (TGFβ) expression, and the mice developed increased airways hyperresponsiveness and airways remodeling [38]. Notably, TGFBR1 and SMAD3, which mediate TGFβ signalling, are susceptibility loci identified in GWAS for asthma [39]. ERBB2 mediates epithelial repair processes, and this function may be defective in subjects with asthma and co-opted by type 2 inflammatory processes [40,41]. Together, these data highlight the complexity of the 17q region, which likely contains multiple independent associations with asthma, variations across asthma endotypes and ethnic groups, and functional variations between blood and airway cells.

Bӧnnelykke et al. [42] performed a GWAS that focused on a severe asthmatic phenotype in children (2-6yrs) characterised by recurrent severe exacerbations requiring hospitalisation, thereby avoiding the need for large sample sizes often required for more heterogeneous phenotypes [43]. Employing 1173 cases and 2522 controls conferred sufficient power to detect associations with the 17q locus, interleukin-33 (IL-33), and a novel signal in the Cadherin-Related Family Member 3 (CDHR3) gene [42]. Subsequently, Bockhov et al. [44] used microarray profiling to identify differentially expressed genes between epithelial cells which were susceptible versus resistant to HRV-C infection, identifying several membrane proteins (e.g., CCRL1, IL-5Rα, and CDHR3) that they hypothesized were potential candidates for HRV-C binding. Ectopic expression of CDHR3 but not the other candidates in HeLa cells facilitated replication of an HRV-C-GFP reporter virus, and structural analysis identified putative binding sites for viral surface proteins which are conserved across all HRV-C subtypes [44]. Notably, the SNP identified by Bӧnnelykke and colleagues (Cys529→Tyr, rs6967330) conferred approximately 10-fold greater HRV-C binding and a 7-fold increase in the number of infected cells [42,44,45]. Moreover, CDHR3 is highly expressed in differentiated ciliated epithelial cells, and cells that are homozygous for the asthma risk variant rs6967330 have 10-fold higher expression levels of the transcription factor FoxJ1, a master regulator of ciliogenesis, which in turn accelerates the development of functional cilia [41,45]. This is an elegant example of how GWAS combined with functional analyses can unlock the mechanisms that determine disease risk.

Exome sequencing is a promising approach to identify rare variants associated with disease, which would fall below the coverage offered by GWAS. Salas et al. [46] sequenced the exome of 54 paediatric cases (<3 years) with severe HRSV infection requiring hospitalisation. They found rare variants in olfactory receptor- and mucin-related genes (OR8U1/8, and MUC6, respectively) [46,47,48]. Further, common disease-associated variants (MAF > 5%) were detected in a similar complement of genes (e.g., OR13C5 and MUC4) as well as genes involved in antigen presentation (HLA-DQA1, HLA-DPB1) [46]. Asgari et al. [49] assessed infection susceptibility in infants (>1 year) hospitalised with severe viral respiratory infections, for which HRSV (56%) and HRV (26%) were the most common pathogens identified. Exome sequencing revealed three rare loss-of-function variants in IFIH1 (MAF < 0.7%), which encodes key viral RNA-sensing receptor melanoma differentiation-associated gene 5 (MDA5) [50]. Mutant IFIH1 isoforms were associated with reduced interferon-β (IFNβ) production and impaired ATPase activity, and only proper function of IFIH1 restricted HRV and HRSV replication [49]. MDA5, a member of the RIG-I-like receptor family, binds viral nucleic acid ultimately leading to the ATPase-dependent transcription of type I interferon genes [51,52].

Whilst GWAS have furthered our understanding of the role of HRV in childhood asthma, there are limitations that are noteworthy. First, it is not possible to dissect complex disease mechanisms employing a single omics modality. Second, susceptibility regions may contain multiple susceptibility genes with pleiotropic functions. Third, GWAS requires massive sample sizes to detect a large number of candidate genes, but the signals identified to date only account for a tiny fraction of the total predicted genetic variance. To address this issue, we would argue that novel perspectives and conceptual advances are required to advance the field. Kitano and Oda first argued that the innate immune systems is governed by a Bow-tie architecture [53]. As illustrated in Figure 1, in a Bow-tie structure, diverse input signals converge on a highly conserved core, which consists of a few non-redundant components, and this fans-out into diverse output signals. We previously proposed that genetic perturbation of the numerous input and output signals would create a lot of noise and complexity, but ultimately the whole system is controlled by a few core pathways [54]. This concept has recently been extended to biological networks more generally into a new conceptual framework to understand the underlying genetic architecture of complex phenotypes. The “omnigenic” model proposes that a given complex phenotypes is dictated by; (1) a minority contribution of variation by a small number of high-penetrance (“core”) genes, and (2) a majority contribution of variation by a large number of low-penetrance (“peripheral”) genes [55]. Under this model, core genes have a direct effect on disease outcome. However, disease risk is driven by a large number of indirect effects from peripheral genes, acting within a highly interconnected regulatory network [55]. We would also like to make the point that the core genes need not be polymorphic. Adoption of the concept of the Bow-tie architecture and the omnigenic inheritance model has potential to transform our understanding of the genetic architecture of complex inflammatory diseases.

## 3. Epigenome

The above studies highlight the utility of GWAS to identify novel susceptibility loci. However, as genetic variation remains constant throughout life and does not vary between cell types, it can be difficult to pinpoint the timing and the cellular context of genetic effects. Epigenetic mechanisms, on the other hand, are dynamic and programmed by environmental cues. Epigenetic mechanisms are active prior to birth, enabling a unique opportunity to track the timing and trajectory of the early origins of disease [56,57]. An epigenome-wide study of DNA methylation patterns in cord blood mononuclear cells (CBMCs) from subjects enrolled in a prospective birth cohort (n = 36) found 589 differentially methylated regions between children who did or did not develop asthma by age 9 [58]. Construction of a molecular interaction network of the genes associated with the differentially methylated regions revealed clustering around regulatory (SMAD3) and pro-inflammatory (IL-1β) gene networks [59,60]. Further, SMAD3 was hypermethylated in the cord blood of asthmatics compared to non-asthmatics, specifically among children born to asthmatic mothers [58]. This finding was replicated in two independent cohorts and was not associated with cell type populations. DeVries et al. [58] also demonstrated increased IL-1β following LPS stimulation of CBMC of children born to asthmatic mother who will later develop asthma compared to non-asthmatics (independent of maternal asthma). As noted above, SMAD3 is a critical mediator of TGFβ signalling, which plays central roles in the regulation of immune responses and also in airways remodeling. Importantly, the finding that SMAD3 methylation patterns are increased at birth in children who develop asthma at age 9 suggests that progression towards asthma may begin in utero [58].

The findings from large-scale GWAS for asthma and related traits suggest that the genome comprises at least 100 susceptibility loci, and that the bulk of these signals are located in non-coding regions of the genome [39,61]. By overlaying these GWAS signals with epigenomic data derived from a broad range of cell types, it is possible to identify cell populations that have open chromatin states in these regions. Employing this approach, multiple studies have demonstrated that candidate genes located near asthma risk loci are preferentially expressed in blood and lung tissue, and are strongly co-localised with tissue-specific regulatory regions, such as enhancers, in immune cells, especially CD4 T cells [27,61,62]. However, it is noteworthy that the reference epigenomic data sets have limited coverage with respect to cell subpopulations (e.g., plasmacytoid dendritic cells) and molecular states (i.e., resting versus activated).

Epigenetic studies of nasal epithelial cells from asthmatic children have observed differential methylation in the HRV receptors CDHR3 and LDLR [63,64]. Further, HRV-infected nasal epithelial cells from children (6–18 years) with asthma identified differential DNA methylation in 16 CpG, including in genes involved in the host immune response against viral infections, such as BAT3 and NEU1 [65]. Risk of severe lower respiratory tract infections, including HRSV, has been associated with increased DNA methylation at birth in the enhancer region of PRF1, which encodes perforin-1, an important mediator of CD8^+^ T cell and NK T cell-mediated cytotoxicity [66,67,68]. DNA methylation of the enhancer region of PRF1 was also observed in children (3–4 years) who have a history of severe HRSV infection compared to healthy controls [69]. Further, studies in experimental mouse models have demonstrated that epigenetic control of histones is important during HRSV infection. In HRSV-infected murine dendritic cells, KDM5B demethylase alterations of H3K4 methylation results in a dampened pro-inflammatory immune response, and an increased Th2 response [70]. In addition, methylation of H3K4 by the histone methyltransferase SMYD3 in regulatory T cells appears important for proper control of airway inflammation following HRSV infection [71]. In human bronchial epithelial cells, IFNγ-dependent changes in histone methylation of H3K4 of the promotor of RIG-I, a pattern recognition receptor that detects viral double-stranded RNA, results in upregulation of RIG-I and increased HRSV clearance [72]. Furthermore, inhibiting histone deacetylase in HRSV-infected airway epithelial cells suppresses HRSV infection and may alleviate virus-induced airway inflammation [73]. Together, these findings demonstrate that epigenetic regulation of key pathways involved in viral sensing (input signals, Figure 1) and immune effector function (output signals) influence risk for severe viral illness in children.

## 4. Transcriptome

Type I and III interferons induce a robust antiviral state in infected and surrounding cells and are essential for immunity to many viruses [74,75]. The dominant paradigm in the literature proposes that type I and III interferons responses to HRV are deficient in subjects with asthma, providing a plausible explanation for the role of these viruses in exacerbations [76]. However, this finding has not been consistently observed in every study, and most studies were performed using samples from adults [76]. The Childhood Asthma Study (CAS) is a prospective birth cohort of children at high risk for asthma, which tracked all episodes of respiratory viral infection over the first 5 years of life [77]. Production of 17 interferon subtypes was assessed at the mRNA level in cord blood mononuclear cells after stimulation with poly-IC. Deficient production of type I and type III but not type II interferons (IFNγ) was observed in samples from 24% of the children, and this was associated with febrile lower respiratory tract infections in the first year of life and increased risk for development of persistent wheeze at age 5 [77]. Notably, by four years of age, type I and type III interferon responses were exaggerated in children who experienced febrile respiratory tract infections in the first year of life compared to children with wheezy infections, suggesting that deficient production of interferons in this subgroup was restricted to a specific temporal window in early infancy [77].

A subset of infants are highly susceptible to acute viral bronchiolitis, and susceptibility to this illness is inversely related to post-natal age. We reasoned that the mechanisms that underpin the heightened susceptibility in infants could be elucidated by comparing immune response patterns during acute viral bronchiolitis in infants versus older children. We collected nasal scrapings and PBMC from children who were hospitalised with acute viral bronchiolitis and stratified the subjects into two age groups—infants (<18 months, n = 15) and children (18 months–5 years, n = 16) [78]. Follow-up samples were collected at post-convalescence. HRSV was detected in 52% of the infants and 7% of the children, whereas detection rates for HRV were 42% and 59% in infants and children, respectively. In PBMC, response patterns in infants were characterised by upregulation of type I interferon-mediated antiviral responses. In contrast, response patterns in PBMC from older children were characterised by upregulation of immunoregulatory and growth factor signalling pathways (PTGER2, TGFβ, ERBB2, VEGF, IL-4, AREG, and HGF) and NK cell cytotoxicity. In the nasal mucosa, responses in both infants and children were dominated by type I and III interferons, but the responses were much more intense in the infants. This intuitive picture of age-dependent differences in host immune responses can become somewhat distorted in the presence of high within-group variability. To address this issue, we employed a personalised analytical approach called N-of-1 pathways, which identifies dysregulated pathways within each individual subject, and can unmask covert immunophenotypes. Two major phenotypes underlying acute viral bronchiolitis responses were discernible in PBMC—one phenotype demonstrated robust interferon responses, while the other displayed a general dampening of interferon and innate immunity and upregulation of type 2 immunity and growth factor signalling [78]. Importantly, the emergent phenotypes were not restricted by age, although the hyperresponsive phenotype was enriched within infants. These data suggest for the first time that the pathogenesis of acute viral bronchiolitis may be driven by distinct immunophenotypes [79].

Weighted gene co-expression network analysis (WGCNA) of gene expression profiles derived from nasal wash/swab samples is a powerful and unbiased technique to reconstruct the molecular networks that are mobilised during respiratory viral infections in children. Employing this approach, we reported that IRF7, a master regulator of type I and III interferon responses [74,75], was a major hub, linking interferon-mediated antiviral responses in asthmatic children with mild-moderate exacerbations [80]. To elucidate the role of IRF7 in HRV responses, we employed siRNA-mediated gene silencing to knockdown IRF7 in airway epithelial cells. The data showed that knockdown of IRF7 reduced the innate antiviral response to HRV infection and increased the expression of proinflammatory mediators (e.g., CXCL5, IL-33, and IL1RL1 [81]). The role of IRF7 has also been investigated in an experimental mouse model of severe viral bronchiolitis. In this model, viral loads were markedly elevated in IRF7 deficient mice, and this unleased the alarmins IL-33 and HMBG1, which induced type 2 inflammation and airways remodeling [82]. Together, these data suggest that IRF7 plays a dual role in respiratory viral infections, by promoting interferon-mediated antiviral responses and limiting the activation of alarmins and type 2 inflammation.

To explore the role of IRF7 gene networks in children with more severe disease, we profiled gene expression in nasal swab samples from children (<17 years old) who presented to the emergency department with severe exacerbations of asthma or wheeze [83]. Hierarchical clustering delineated two major molecular phenotypes of HRV-induced wheeze, which were characterised by IRF7hi versus IRF7lo gene network patterns. Children with the IRF7hi molecular phenotype were characterised by robust upregulation of type I interferon responses, whereas the IRF7lo phenotype lacked an IRF7 signature and instead were characterised by upregulation of Type 2 inflammation (IL-4R, FCER1G, ARG1, and SERPINB3) and growth factor signalling pathways (TGFβ, CSF3, and EGF) and downregulation of interferon-γ. Importantly, the IRF7 phenotypes were associated with distinct clinical features; IRF7hi children presented to emergency approximately two days after the onset of clinical symptoms, whereas IRF7lo children presented approximately 5 days post-first symptoms (Figure 2). The IRF7lo phenotype was also associated with increased risk of admission to hospital and a shorter time to recurrence [83].

Altman et al. [84] collected nasal lavage samples from asthmatic children at baseline and at two timepoints (0–3 days, 4–6 days) after the onset of cold symptoms. WGCNA was employed to elucidate gene network patterns and their dynamic states during viral and non-viral exacerbations. Expression of a type I interferon response module (IRF7, STAT1, and STAT2) was upregulated in virus positive subjects; expression of the module peaked at approximately 2 days after cold onset and the expression intensity was increased in children with exacerbations compared to those without exacerbations. Exacerbations responses were also associated with initial (0–3 days) enhanced epithelial-associated SMAD3 signalling and downregulation of lymphocyte-related pathways, followed by later upregulation of EGFR signalling [85,86], mucus hypersecretion [87], and eosinophil activation [88,89]. Finally, the authors demonstrated that the ratio of expression between a type 2 inflammatory module and the type I interferon response module at baseline could predict time to recurrence.

## 5. Microbiome

It has been known for some time that children who are raised in traditional farming environments are protected from the development of allergies and asthma [90]. To elucidate the underlying mechanisms, Stein et al. [91] conducted an elegant study to compare microbial exposure and innate immunity in Amish and Hutterite farm children. Notably, the Amish of Indiana and the Hutterites of South Dakota have similar diets, lifestyles, and genetic backgrounds. However, the Amish employ traditional farming methods and live on single-family farms in close proximity to their animals. In contrast, the Hutterites live in large communal farms, and the animals are housed in large industrial complexes away from their homes. Strikingly, the prevalence of asthma and allergies is approximately 4-5-fold lower in Amish children compared to Hutterite children. House dust samples collected from Amish homes differ from those of Hutterite households, most notably with 7-fold higher levels of endotoxin (lipopolysaccharide (LPS)) [91]. In peripheral blood, proportions of neutrophils were increased, and proportions of eosinophils were decreased in Amish compared to Hutterite children. Moreover, Amish and Hutterite children displayed distinct gene network patterns in peripheral blood leukocytes, which were characterised by upregulation of TNF- and IRF7-associated gene networks in the Amish [91]. Finally, exposure of dust collected from Amish but not Hutterite households was sufficient to inhibit the development of asthma-related traits in an experimental mouse model, and this protection was dependent on the core innate immune signalling hubs MyD88 and TRIF [91,92]. Together, these data demonstrate that exposure to microbial products in early life can rewire innate immune programs and prevent the development of allergy and asthma.

The protective “farm effect” is not restricted to farming practices per se, but rather is specifically determined by the microbial composition present. Kirjavainen et al. [93] reasoned that protective indoor microbiota from farm houses should confer similar protection in non-farm houses, irrespective of environmental and lifestyle factors. Indeed, exposure to farm-like microbial relative abundance at age two months was associated with decreased risk of asthma development by six years of age in individuals who grew up in farm and non-farm households. Thus, microbial compositional exposure in early life is a readily identifiable predictor of asthma development and therefore represents a modifiable intervention strategy [93].

The host microbiome is a crucial determinant of innate immune maturation in early life. Arrieta et al. [94] found that reduced relative abundance of four bacterial genera in the gut microbiome in the first 100 days of life marked infants at risk of asthma. Decreased relative abundance of these bacteria was associated with reduced LPS (endotoxin) biosynthesis within the microbiome. To further explore this finding, adult germ-free mice were inoculated with human faecal microbiota, with or without supplementation with the identified protective species. Among the subsequent generation, mice born to supplemented mothers maintained the protective gut bacteria and exhibited lower airway inflammation following experimental challenge, characterised by reduced airway infiltration of immune cells and dampened pro-inflammatory cytokine production [94]. The findings demonstrate that maternal exposure to specific microbes and their products can modulate risk of the development of asthma in the offspring.

The airway microbiome is highly dynamic in early life, and matures to a stable, diverse profile later than microbial communities from other sites [95,96]. In the CAS study, Teo et al. [97] examined the development of the infant (n = 234) nasopharyngeal microbiome over the first year of life, capturing data during predefined asymptomatic periods and all symptomatic respiratory viral episodes. *Streptococcus*-, *Moraxella*-, and *Haemophilus*-dominated profiles were all significantly more frequent during acute respiratory infections compared to asymptomatic periods and conferred a higher risk of more severe illness (presence of fever and/or infection spread to the lower airways) [97]. In addition, high abundance *streptococcus* colonisation was significantly more frequent in infants who developed wheeze at age 5. At 5 years of age, these bacteria maintained their dominant colonisation and association with greater risk of respiratory viral infection and chronic wheeze, particularly alongside allergic sensitisation [98]. Importantly, the shift towards increased abundance of these bacteria frequently *preceded* viral detection and the onset of symptoms, suggesting that transient incursions of the airway microbiome with pathogenic microorganisms may destabilise homeostatic mechanisms and increase exacerbation risk [98,99]. The post-infection recovery of airway microbial populations appears important for future asthma risk. A multi-centre prospective cohort of infants (<1 year, n = 842) hospitalised with bronchiolitis found that participants with increased airway colonisation with *Moraxella* and *Streptococcus* after viral clearance (3 weeks post-hospitalisation) were at increased risk for recurrent wheeze at 3 years of age [100]. This was independent of the causative agent of viral bronchiolitis and no association was seen from index samples at the time of hospitalisation. *Moraxella* and *Streptococcus* genera were also prominent in the nasal microbiomes of older asthmatic children (6–17 years, n = 413) followed over the fall season [101]. *Moraxella*-dominated microbiomes were prevalent among asthmatic children and were associated with increased risk of exacerbation and eosinophil activation, particularly in the younger participants. Furthermore, in vitro inoculation of epithelial cells with *M. catarrhalis* provoked greater epithelial damage and inflammatory cytokine production [101]. Additionally, asthmatic airway microbiomes dominated by *Streptococcal* species was associated with increased risk of HRV infection over the virus season. Further studies investigating acute bronchiolitis have found an association of these bacterial species with the severity of HRSV infections and identified differences in airway microbial composition between HRSV and HRV infections [102,103,104,105].

Taken together, exposure to microbes and their products can confer both risk or protection from the development of asthma-related traits and the triggering of exacerbations, depending on the timing, the location, and the specific microorganisms involved. Accordingly, administration of specific microbes or their products in utero or in early infancy is a plausible strategy to reprogram innate immunity for the primary prevention of asthma and wheeze [28,106,107,108].

## 6. Multi-Omics

Multi-omic investigations provide unique insight into disease mechanics as they simultaneously assess the joint contribution of molecular features across multiple layers of biological regulation on a disease process or phenotype, and unveil emergent properties of biological systems. Consequently, multi-omics studies have increased predictive power over single-omics analysis [109,110]. For example, Zhou et al. [111] demonstrated improved performance in classifying respiratory viral infection events from baseline when the classifier was built from the integration of multi-omic data. A key challenge of multi-omic research is the availability of computational tools to synthesise data from different omics platforms that are heterogeneous with respect to the scale, noise, and inherent bias of each platform. Data integration methods are generally divided into two subgroups; integration of data derived from different cohorts of subjects using the same omics platform (Horizontal), or alternatively integration of data derived from different omics modalities on the same subjects (Vertical) [112,113,114]. Vertical integration methods can be further divided into parallel or hierarchical methods; the former approach is agnostic to the data type, and the latter approach takes into account the sequence of information exchange in biological systems (e.g., →mRNA→protein) [114]. The application of multi-omic technologies to interrogate immune responses in early life is extremely challenging because there are limitations in the amount of sample that can be collected from infants.

Recently, novel sample processing protocols have been developed which have enabled the partition of very small blood volumes into separate aliquots that are sufficient for multi-omic studies. Olin et al. [115] applied a systems biology approach to investigate immune development over the first 3 months of life in pre-term and term births, from as little as 100 µL of blood. Targeted serum proteomics and whole blood mass cytometry revealed striking differences in immune cell populations between cord blood and peripheral blood collected from newborns in the first week of life [115]. Further, integrated topological analysis revealed distinct clustering according to term/pre-term birth status in newborns. Importantly, immune cell and proteome profiles from week 1 of life preterm and term participants converged to a shared trajectory within weeks, which was particularly evident in neutrophils and naïve CD4^+^ T cells [115]. In contrast, transcriptomic analysis of a subset of individuals revealed gene transcription regulation was not equivalent at 3 months between the individuals from term and preterm births, indicating that immune function may be altered even though cellular profiles have converged. Additionally, faecal microbiome analysis generally displayed increasing diversity over the first 3 months of life, although individuals who exhibited early dysbiosis were characterised by perturbed developmental immune trajectory. Taken together, these data suggest that drastically different immune profiles at birth may converge to a stereotypic trajectory determined by a (sufficiently) large number of response-eliciting immune exposures [115].

Lee et al. [116] investigated immune development over the first week of life with multi-omic data generated from >1 ml blood. Transcriptomic, proteomic and metabolomic data, as well as cellular immunophenotyping and cytokine profiling, demonstrated dramatic biological change over the first week of life compared to baseline (day of life 0). Three separate integration strategies were employed; prior knowledge-based network construction (NetworkAnalyst [117]), multivariate biomarker detection (DIABLO [118]), and multiscale, multifactorial response network (MMRN [119]). The integration results did not overlap at the level of features/variables, although they did exhibit common biological themes from higher-level functional pathways, most notably type I interferon and neutrophil signalling and complement activation. Furthermore, these findings were validated in an independent cohort, suggesting immune development over the first week of life is not random but follows a robust, shared trajectory, irrespective of interindividual variation [116].

Together, these studies have revealed that the neonatal immune system undergoes rapid developmental changes during the first few weeks of life. Upregulation of key innate immune defense functions exemplified by the type I interferon pathway would presumably provide protection from respiratory viral infections during this crucial period of heightened susceptibility.

## 7. Single-Cell Omics

Immune function relies on the coordinated activity of multiple populations of innate and adaptive immune cells, with surrounding structural cells in the local tissue microenvironment. The development of single-cell profiling enables the first dissection of complex biological processes at single-cell resolution. A promising application of scRNAseq is to establish cellular atlases which deeply profile blood/tissue in healthy versus disease states to provide a reference framework for future research [120,121,122,123,124,125]. For example, single-cell genomics led to the discovery of the pulmonary ionocyte and its role as the primary source of CFTR gene activity in the airway epithelia, which is important for understanding the pathogenesis of cystic fibrosis [126,127]. Villiani et al. [128] employed scRNAseq to profile monocytes and dendritic cell (DCs) in healthy adult PBMC, defining a revised taxonomy of these critical cells in innate immunity. Notably, identification of a novel DC subset, distinguished by AXL and SIGLEC expression, demonstrated homology to both conventional DCs and plasmacytoid DCs [128]. In addition to the discovery of novel cell types and states, because cells are continuously undergoing changes in their functional state, and this process is not synchronised between cells, it is possible to extract dynamic information from single-cell profiles and project cells onto temporal (pseudotime) or developmental trajectories [129].

During infections, dynamic interactions between infected cells, immune cells and other bystander cells are needed to clear the infection and restore normal homeostasis. Ligands produced by “sender cells” are recognised by receptors on “receiver cells”, triggering downstream signalling events, which ultimately activates target genes. Computational methods have been developed that leverage ligand–receptor expression patterns between cells to infer cell-to-cell communication networks that mediate immune activation [130]. 

Krausgruber et al. [131] employed RNA-Seq and ATAC-Seq to elucidate the role of three structural cell populations (endothelial, epithelial, fibroblasts) across 12 tissues in regulation of the immune response to viral infection. Notably, by inferring cell-to-cell communication networks on the basis of ligand–receptor pairs, they identified substantial cross talk between structural and immune cells. Moreover, there were substantial differences between immune gene profiles from the same structural cell types across different organs, indicating an underappreciated role of structural cells in the regulation organ-specific immune responses. Importantly, integrative analysis of epigenetic and transcriptomic data revealed an epigenetic immune potential in structural cells, defined by gene expressed at low levels during homeostasis which were “epigenetically poised” for rapid upregulation [131]. Indeed, functional evaluation employing an in vivo viral challenge model demonstrated preferential activation of genes with unrealized potential, suggesting structural cells are epigenetically pre-programmed to rapidly respond to viral exposure. These findings highlight the potential for single-cell genomic approaches to generate expression atlases across asthma-relevant tissues and contexts, and leverage ligand–receptor network analyses to decipher immune/structural cross talk in an organ-specific and organism-wide manner.

## 8. Conclusions

HRSV and HRV are the most frequent agents of respiratory infection in infants and children and are major triggers of bronchiolitis and wheeze/asthma exacerbations, placing an enormous burden of global health care resources. The Bow-tie architecture and omnigenic inheritance model represents an important conceptual advance that posits that a small number of core pathways drive disease processes through many input and output signals, which themselves contribute to disease risk through direct and indirect signalling paths [54,55]. Omics investigation have begun to unveil the molecular details and intricate signalling pathways that govern the innate immune responses to respiratory viruses. Assessment of gene expression profiles in children with severe viral-induced exacerbations of asthma/wheeze uncovered two distinct molecular phenotypes—one defined by high expression of IRF7 and a robust interferon/antiviral response and the other characterised by low IRF7 expression and upregulation of markers of TGFβ signalling and type 2 inflammation [83]. Comparable phenotypes are also evident in infants and children who were hospitalised with acute viral bronchiolitis [78]. The first asthma GWAS identified the 17q locus as a major susceptibility locus for asthma, which is specifically associated with early-onset asthma in children who wheeze with HRV infection [29]. Subsequent GWAS identified further candidates in diverse regions across the genome [39,42,61]. Importantly, functional validation of GWAS candidate asthma risk genes have uncovered key roles in HRV infection susceptibility (ORMDL3 and CDHR3) and TGFβ signalling (GSDMB, TGFBR1, and SMAD3) [34,35,39,44]. Furthermore, a subset of individuals display deficient type I and III interferon responses at birth and this confers greater risk of developing febrile lower respiratory tract infections in the first year of life and persistent wheeze in early childhood [77], indicating that the trajectory towards asthma begins at birth or in utero. In the first week of life, expression of the interferon system is rapidly upregulated and by inference the innate immune system is poised to respond to respiratory viral infections [116], and in this context it is noteworthy that infants with severe viral bronchiolitis displayed exaggerated type I and III interferon responses in the airways compared to older children [78]. Finally, early life exposures to microbes and their products are critical for training appropriate innate immune responses, and can confer protection from development of allergies and asthma in children [91]. Therefore, administration of microbial products in early life is a promising strategy for the primary prevention of asthma and wheeze.

## Figures and Tables

**Figure 1 ijms-21-06372-f001:**
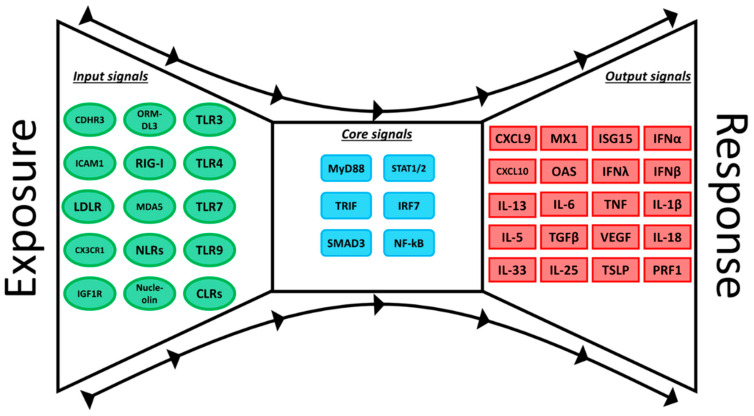
The innate immune system is governed by a Bow-tie architecture. The Bow-tie structure enables diverse input signals to converge on a few core pathways, which in turn drive functional responses through the actions of a large number of effector and regulatory molecules. We have illustrated this concept by providing specific examples of key molecules associated with host responses to HRSV, HRV, and asthma risk.

**Figure 2 ijms-21-06372-f002:**
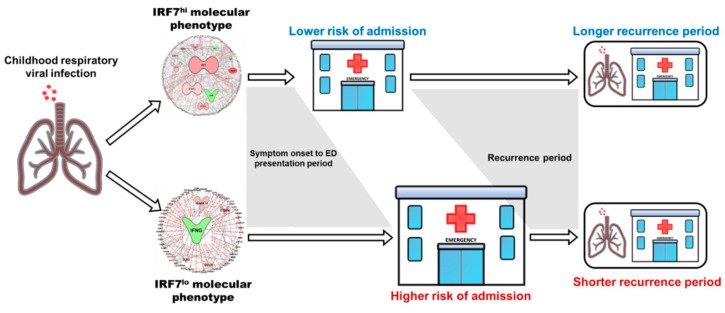
Schematic representation of outcomes associated with IRF7hi and IRF7lo molecular phenotypes underlying severe exacerbations of asthma and wheeze. Gene network diagrams reproduced with permission; originally published in *The Journal of Immunology* [83], copyright © 2020 The American Association of Immunologists, Inc.

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
