# Peer review of "Decoding Susceptibility to Respiratory Viral Infections and Asthma Inception in Children"

_ijms, 2020, doi:10.3390/ijms21176372_

Round 1

Reviewer 1 Report

X

Manuscript ID: ijms-901960

This review discusses the application of current tools to understand the contribution of human respiratory syncytial virus (HRSV) and human rhinovirus (HRV) in asthma. However, the title is misleading. References are missing. Specific comments are as follows:

  1. The title does not reflect the focus of this review on genetics and asthma, that needs to be modified.
  2. Figure 1. It is not clear how this figure has been constructed. It indicates the bow-tie model is related to “the innate immune response to respiratory viral infection”. However, it implies that it applies to any viral infection, when that is not the case since viral infections activate/inhibit genes differentially. That needs to be clarified or the figure needs to be corrected. It needs to better indicate how that related to asthma and HRV and HRSV.
  3. The authors need to include the differences of the type of asthma linked to either virus HRV or HRSV. Atopic vs non-atopic, recurrent wheeze, etc. The review would benefit from a brief description of the clinical manifestations of asthma developed by either virus.
  4. According to the most recent ICTV report, Human respiratory syncytial virus is abbreviated HRSV. That needs to be corrected throughout the text.
  5. There are some published reviews focused on HRSV, HRV, asthma and the immune response. The difference from this review may reside on its focus of the application of the different tools to dissect the underlying mechanism. That needs to be reflected in the abstract and title.
  6. Some references for HRSV and asthma should be included in this work (e.g. from the groups of L. Bont, O. Ramilo).

Author Response

We would like to thank the reviewers for their time and expertise in reviewing our manuscript. We have revised the manuscript to address the comments and feedback we received, and we hope that it is now suitable for publication.

Reviewer 1: The title does not reflect the focus of this review on genetics and asthma, that needs to be modified.

Author response: We have amended the title: “Decoding susceptibility to respiratory viral infections and asthma inception in children”

Reviewer 1: Figure 1. It is not clear how this figure has been constructed. It indicates the bow-tie model is related to “the innate immune response to respiratory viral infection”. However, it implies that it applies to any viral infection, when that is not the case since viral infections activate/inhibit genes differentially. That needs to be clarified or the figure needs to be corrected. It needs to better indicate how that related to asthma and HRV and HRSV.

Author response: We would like to clarify that the concept of the Bow-tie architecture refers to the organisation of the innate immune system more generally. We have updated the figure legend to reflect this point and provide examples of specific pathways associated with HSRV, HRV, and asthma risk.

“Figure 1. The innate immune system is governed by a Bow-tie architecture.  The Bow-tie structure enables diverse input signals to converge on a few core pathways, which in turn drive functional responses through the actions of a large number of effector and regulatory molecules. We have illustrated this concept by providing specific examples of key molecules associated with host responses to HRSV, HRV, and asthma risk.”

Reviewer 1: The authors need to include the differences of the type of asthma linked to either virus HRV or HRSV. Atopic vs non-atopic, recurrent wheeze, etc. The review would benefit from a brief description of the clinical manifestations of asthma developed by either virus.

Author response: To address this issue, we have amended the introduction as follows (lines 48-62): 

“What remains an open question is whether or not these viral agents drive the development of the various clinical manifestations of asthma and wheeze (e.g. atopic asthma, non-atopic asthma, transient/persistent/recurrent wheezing) in susceptible individuals or alternatively unmask a pre-existing susceptibility in children who were already on a trajectory towards asthma. Whilst both HRSV and HRV induce wheezing, HRV wheezing is a much stronger predictor of subsequent asthma than HRSV wheezing, especially in children with aeroallergen sensitization [12]. Moreover, prevention of RSV infection in pre-term or high-risk infants with the monoclonal antibody palivizumab decreased recurrent wheezing but had no effect on atopic asthma or lung function [14-17]. Oral prednisolone treatment of first-time wheezers with HRV decreased time to recurrence in the subgroup of children with high viral loads. Together, these data suggest that HRSV is a risk factor for non-atopic wheeze/asthma whereas HRV drives atopic asthma. Given that asthma is a highly complex and heterogeneous disease, untangling the role of these viruses in the development of specific asthma phenotypes will require a detailed understanding of the underlying cellular and molecular mechanisms.”

Reviewer 1: According to the most recent ICTV report, Human respiratory syncytial virus is abbreviated HRSV. That needs to be corrected throughout the text.

Author response: We have amended the abbreviation of “RSV” to “HRSV” throughout the manuscript.

Reviewer 1: There are some published reviews focused on HRSV, HRV, asthma and the immune response. The difference from this review may reside on its focus of the application of the different tools to dissect the underlying mechanism. That needs to be reflected in the abstract and title.

Author response: As noted above, we have amended the title. We also included the following sentence in the abstract (lines 13-15) and removed the final sentence of the Introduction.

“Here, we will discuss the application of the powerful tools of systems biology to decode the molecular mechanisms that determine risk for infection and subsequent asthma”

Reviewer 1: Some references for HRSV and asthma should be included in this work (e.g. from the groups of L. Bont, O. Ramilo).

Author response: We have included the following two references:

  1. Mejias, A.; Wu, B.; Tandon, N.; Chow, W.; Varma, R.; Franco, E.; Ramilo, O. Risk of childhood wheeze and asthma after respiratory syncytial virus infection in full-term infants. Pediatr Allergy Immunol 2020, 31, 47-56, doi:10.1111/pai.13131.

  1. Scheltema, N.M.; Nibbelke, E.E.; Pouw, J.; Blanken, M.O.; Rovers, M.M.; Naaktgeboren, C.A.; Mazur, N.I.; Wildenbeest, J.G.; van der Ent, C.K.; Bont, L.J. Respiratory syncytial virus prevention and asthma in healthy preterm infants: a randomised controlled trial. Lancet Respir Med 2018, 6, 257-264, doi:10.1016/S2213-2600(18)30055-9.

Reviewer 2 Report

Read and Bosco thoroughly review the current understanding of asthma derived from genetic and molecular profiling studies and its links to bronchiolitis caused by respiratory syncytial virus (RSV) and/or rhinovirus (HRV) infection. Both have been linked to wheezing and subsequent asthma in the initiation and/or the exacerbation phase. The central message is the “Bow-Tie” response that has many sensors, a few transmitters and many effectors.

The review is very well written, clear, comprehensive and informative. It is an excellent synthesis of results from many, disparate approaches all converging on asthma.

The only topic that was not addressed was which virus is the initiator for wheezing, RSV or HRV. Clearly both can be inducers of exacerbation.

The central focus and organizing principle of this review is the mediators of the host response encapsulated nicely in the “bow-tie” model. In the legend to Fig. 1 and l.153, “Bow” is capitalized but usually not. It seems important enough to be “Bow-tie” or even “Bow-Tie” everywhere.

In several places “=” in the size of the study (n=19) is highlighted in yellow (l.245 and beyond). It is not clear why, but it needs to be removed.

l.355. developed

l.436. enables the first dissection of…

Author Response

We would like to thank the reviewers for their time and expertise in reviewing our manuscript. We have revised the manuscript to address the comments and feedback we received, and we hope that it is now suitable for publication.

Reviewer 2: Read and Bosco thoroughly review the current understanding of asthma derived from genetic and molecular profiling studies and its links to bronchiolitis caused by respiratory syncytial virus (RSV) and/or rhinovirus (HRV) infection. Both have been linked to wheezing and subsequent asthma in the initiation and/or the exacerbation phase. The central message is the “Bow-Tie” response that has many sensors, a few transmitters and many effectors.

The review is very well written, clear, comprehensive and informative. It is an excellent synthesis of results from many, disparate approaches all converging on asthma.

Author response: We thank the reviewer for their positive comments.

Reviewer 2: The only topic that was not addressed was which virus is the initiator for wheezing, RSV or HRV. Clearly both can be inducers of exacerbation.

Author response: To address this issue, we have amended the introduction as follows (lines 48-62): 

“What remains an open question is whether or not these viral agents drive the development of the various clinical manifestations of asthma and wheeze (e.g. atopic asthma, non-atopic asthma, transient/persistent/recurrent wheezing) in susceptible individuals or alternatively unmask a pre-existing susceptibility in children who were already on a trajectory towards asthma. Whilst both HRSV and HRV induce wheezing, HRV wheezing is a much stronger predictor of subsequent asthma than HRSV wheezing, especially in children with aeroallergen sensitization [12]. Moreover, prevention of RSV infection in pre-term or high-risk infants with the monoclonal antibody palivizumab decreased recurrent wheezing but had no effect on atopic asthma or lung function [14-17]. Oral prednisolone treatment of first-time wheezers with HRV decreased time to recurrence in the subgroup of children with high viral loads. Together, these data suggest that HRSV is a risk factor for non-atopic wheeze/asthma whereas HRV drives atopic asthma. Given that asthma is a highly complex and heterogeneous disease, untangling the role of these viruses in the development of specific asthma phenotypes will require a detailed understanding of the underlying cellular and molecular mechanisms.”

Reviewer 2: The central focus and organizing principle of this review is the mediators of the host response encapsulated nicely in the “bow-tie” model. In the legend to Fig. 1 and l.153, “Bow” is capitalized but usually not. It seems important enough to be “Bow-tie” or even “Bow-Tie” everywhere.

Author response: The term “Bow-tie” has now been capitalised throughout the manuscript.

Reviewer 2: In several places “=” in the size of the study (n=19) is highlighted in yellow (l.245 and beyond). It is not clear why, but it needs to be removed.

Author response: We have removed the yellow highlighting.

Reviewer 2: l.355. developed

l.436. enables the first dissection of…

Author response:

I.373  “development” was changed to “developed”.

l.451 “… enables for the first the dissection of …” was changed to “… enables the first dissection of …”